# Revealing Taxi Interaction Network of Urban Functional Area Units in Shenzhen, China

Guijun Lai [1], Yuzhen Shang [1], Binbao He [1], Guanwei Zhao [1,2,*] and Muzhuang Yang [1,2]

1   School of Geography and Remote Sensing, Guangzhou University, Guangzhou 510006, China;
    2111901035@e.gzhu.edu.cn (G.L.); 2112001045@e.gzhu.edu.cn (Y.S.); 1901400031@e.gzhu.edu.cn (B.H.);
    ymz@gzhu.edu.cn (M.Y.)
2   Institute of Land Resources and Coastal Zone, Guangzhou University, Guangzhou 510006, China
*   Correspondence: zhaogw@gzhu.edu.cn

**Abstract:** Characterizing the taxi travel network is of fundamental importance to our understanding of urban mobility, and could provide intellectual support for urban planning, traffic congestion, and even the spread of diseases. However, the research on the interaction network between urban functional area (UFA) units are limited and worthy of notice. Therefore, this study has applied the taxi big data to construct a travel flow network for the exploration of spatial interaction relationships between different UFA units in Shenzhen, China. Our results suggested that taxi travel behavior was more active in UFA units dominated by functions, including residential, commercial, scenic, and greenspace during weekends, while more active in UFA units dominated by industrial function during weekdays. In terms of daily average volume, the characteristics of spatial interaction between the various UFA types during weekdays and weekends were similar. During the morning peak period, the sink areas were mainly distributed in Futian District and Nanshan District, while during the evening peak period, the sink areas were mainly distributed in the southern part of Yantian District, the southwestern part of Longgang District, and the eastern part of Luohu District. The average daily taxi mobility network during weekdays showed a spatial pattern of "dense in the west and north, sparse in the south and east", exhibiting significant spatial unevenness. Compared with weekdays, the daily taxi mobility network during weekends was more dispersed and the differences in node sizes decreased, indicating that taxi travel destinations were more diverse. The pattern of communities was more consistent with the administrative division during weekdays, indicating that taxi trips are predominantly within the districts. Compared with weekdays, the community pattern of network during weekends was clearly different and more in line with the characteristics of a small world network. The findings can provide a better understanding of urban mobility characteristics in Shenzhen, and provide a reference for urban transportation planning and management.

**Keywords:** taxi; big data; complex network; urban functional area; small world; point-of-interest (POI); urban mobility

## 1. Introduction

Urban mobility is a classic and hot issue in the fields of geography, transportation, and urban planning. Spatiotemporal characteristics of urban mobility can explain the homogeneity of the influence of individual residents' behaviors on urban space [1]. Taxis are an important element in the urban transportation system due to their ability to provide flexible, personalized services and cover a wide geographic area [2–6]. Numerous studies have been achieved on the taxi travel behavior from various aspects.

In recent years, two papers published in the journals Nature and Science [7,8] have promoted the application of complex network analysis methods. To date, the study of complex networks has permeated various fields, including sociology, biology, physics, economics, computer science, transportation, etc. As the complex network can provide the

ability to study transportation modes and their interactions [9–11], many scholars have applied complex networks for taxi travel analysis [12–18]. Urban functional area unit is an important indicator to describe the spatial heterogenies of urban functions. Analyzing the traffic flow interactions between various urban functional area units will benefit our understanding of urban mobility and provide a reference for optimizing urban land use. However, most studies analyzed the travel pattern in the scale of regular grid, and the studies from the perspective of urban functional area units are limited. Specifically, how is the taxi interaction network among different UFA units? Answering this question can not only improve our understanding of urban mobility, but also provide a reference for urban transportation planning.

Our study attempts to answer this question from two aspects. First, what is the spatial distribution of UFA units in Shenzhen? Second, how does the taxi flow interact between different UFA units? As one of the central cities in the Guangdong-Hong Kong-Macao Bay Area, Shenzhen has a high population density and a complex urban function structure in the region, and the contradiction between residents' travel demand and urban function structure is particularly prominent. Therefore, we applied the complex network analysis method to study the spatial interaction characteristics of taxi flow between different urban functional area units in Shenzhen. The findings can provide invaluable insights into policy formation regarding taxi travel to guide transport development in Shenzhen, China, and worldwide.

The remainder of the paper is organized as follows. Section 2 includes a literature review on our subject. In Section 3, the study area, data, and methods are described. Section 4 applies taxi trip data to uncover the urban mobility network characteristics in the scale of UFA units. Section 5 includes our conclusions and prospect.

## 2. Literature Review

Since UFA is the basic unit of network analysis in this paper, an overview of the progress in UFA identification is first provided. UFA is one of the important indicators to quantify the spatial structure of a city. Precise identification of UFA units is a prerequisite for understanding the spatial structure of cities, and thus making rational urban plans. For this reason, many scholars have conducted research on this issue. The traditional methods of UFA identification mainly include expert judgment and survey statistics. However, these methods are not suitable for use in the era of big data due to several shortcomings, such as not objective enough, time-consuming, and error-prone. Points of interest, as a common type of big data, have turned into one of the main data sources for UFA identification studies due to their wide coverage, large data volume, and easy access. The identification of UFA units is essentially a process of classification. The classification methods for UFA units mostly include statistical analysis method, kernel density estimation method, topic model method, and cluster analysis method. For example, Long et al. identified the UFA units in Beijing, China using POI data and bus IC card data [19]. Li et al. identified the UFA units in Wuhan, China by applying the kernel density estimation method, and demonstrated the high accuracy by comparison with urban planning map and high-resolution remote sensing images [20]. Chen et al. applied the topic model in Guangzhou, China based on the POI data and taxi trajectory data, and identified several UFA types, such as mature residential areas, commercial and entertainment areas, and development areas [21]. Yan et al. realized the UFA identification of Dongying city, China using a combination of KD-tree clustering algorithm, kernel density algorithm, and image element threshold method, and the Kappa value of result was 0.763 [22]. In summary, previous studies still have shortcomings. For example, due to the characteristics of POI data, different POIs have different social perceptions (for example, in the category of medical facilities POI, the influence of tertiary hospitals and general clinics varies greatly), the studies using only a single method will significantly affect the accuracy of identification results. To this end, we propose a method for UFA identification that integrates the statistical analysis and kernel density estimation methods to improve the accuracy. The main improvement can be concluded in the fact that

different identification methods are used respectively in accordance with the difference between POI types. For cognitive POIs, the frequency density and type ratio calculation methods are used, while for density POIs, the kernel density estimation method is used to reduce the influence of noise.

Next, a brief overview of the progress in taxi travel behavior is presented. As mentioned above, many studies were conducted on taxi travel from different aspects. Taxi travel pattern is one of the classic topics in urban mobility research. For example, Liu et al. explored the taxi travel pattern of Shenzhen, China, and found that people are more prone to visit leisure places during the weekends, and choose to engage in sports and see the doctor both on weekdays and weekends [23]. Tang et al. characterized the urban mobility from taxi trips in Harbin, China, and found the distribution pattern of origins and destinations on weekdays and weekends [24]. Shen et al. investigated the spatial and temporal patterns of taxi trajectory in Nanjing, China, and found that the temporal pattern shows a strong daily rhythm, while the spatial pattern shows that the number of pick-up and drop-off locations gradually diminishes from the downtown areas to the outer suburbs [25]. Few scholars have further analyzed the influencing factors of taxi travel characteristics. For example, Ge et al. claimed that the health care area is the most critical factor in all land-use variables that impact taxi ridership based on the comparison between Shanghai and NYC [4]. Qian et al. found that the urban form has a significant impact on urban taxi ridership. In particular, the accessibility to subways is positively associated with the taxi ridership [5]. Li et al. found that the land-use mix has a positive effect on taxi travel in Chengdu, China [26]. Feng et al. identified the critical roads and intersections based on taxi trip data in Lanzhou, China [27]. In addition, some topics have been explored, such as transportation emission, transportation network recognition, travel demand prediction, the impact of built environment on taxi travel behavior, the relationship between taxis, buses, subway, etc. [4,5,24,28–33]. Since our study focuses on the characteristics of taxi trips among different UFA units, research progress beyond the taxi travel pattern will not be elaborated.

The methods involved in the study of the spatiotemporal characteristics of taxi trips can be summarized into four categories: Spatial statistical methods, machine learning methods, time series methods, and complex network analysis methods. For example, Yue et al. illustrated the hotspot areas of residents' trips using the clustering analysis method [34]. Pan et al. applied the spatiotemporal clustering of taxi pick-up and drop-off locations as a way to discover hotspot areas in cities [35]. Jiang et al. applied the exploratory spatial data analysis method to study the human travel pattern [36]. Gong uncovered the travel pattern of taxi passengers in Shanghai using Bayesian models [37]. Since 1998, complex network analysis methods have started to attract attention from scholars in many fields, such as computer science, sociology, geography, etc. [38]. As the urban transportation network is a typical complex system, it is undoubtedly well suited to apply complex network analysis methods to reveal urban travel characteristics. For example, Liu et al. applied the complex network analysis to reveal the community structure of taxi travel network in Shanghai, China [39]. Xiao et al. found that Shanghai presents a two-level hierarchical polycentric urban structure feature by applying a spatial embedded network model [40]. To build the original destination (OD) matrix, the common types of traffic analysis unit, include regular grids [41–45], hexagons [46], and irregular polygons (usually obtained by dividing the road network or district boundary). In particular, the regular grid is the most common due to its advantages, such as ease of use and visualization. Some studies use irregular polygons as traffic analysis units, which are obtained by overlaying road networks with administrative boundaries. There is no doubt that the usage of road network based traffic analysis unit is more in line with the driving characteristics of taxi and is more conducive to the accurate meaning of taxi mobility network characteristics. However, to the best of the authors' knowledge, little research has been undertaken to consider the urban functional attributes of the traffic analysis unit. To this end, we first applied an improved method for urban functional area identification. Then, we analyzed

the characteristics of Shenzhen taxis travel network at the scale of urban functional area unit, and explored the spatial interaction pattern of various urban functional area types. Our findings are expected to provide reference information for traffic management and urban planning.

## 3. Materials and Methods

### 3.1. Study Area

Shenzhen, as one of China's mega cities and special economic zones, is a highly urbanized city. Shenzhen has nine administrative districts (Futian District, Luohu District, Nanshan District, Baoan District, Longgang District, Yantian District, Longhua District, Pingshan District, Guangming District) and one functional district (Dapeng New District), with 74 subdistricts in total. Shenzhen is located on the east coast of the Pearl River Estuary, bordering with Hong Kong, Dongguan, and Huizhou, with a total area of 1997.47 km$^2$. By the end of 2020, the number of resident population in Shenzhen was 17.56 million [47], and the population density was 8791 persons/km$^2$. It can be inferenced that Shenzhen has the highest population density among large and medium-sized cities in China, and even far exceeds some international metropolises, such as Tokyo (6372 persons/km$^2$ [48]) and London (5701 persons/km$^2$ [49]). Shenzhen is also an important transportation hub city in South China, with a variety of transportation modes, such as airports, railway stations, bus terminals, and sea ports for external travel and convenient public transportation facilities, such as subway, bus, and taxi for internal travel. By the end of 2017, there were 18,379 taxis in Shenzhen. Prior to 5 May 2017, the taxi fares in Shenzhen varied from company to company. The flag-fall prices for red, yellow, and green taxis were CNY 11 per 2 km, CNY 10 per 2 km, and CNY 6 per 1.5 km, respectively during the day, and CNY 16 per 2 km, CNY 13 per 2 km, and CNY 7.2 per 1.5 km, respectively during the night. Since 5 May 2017, the taxi fares in Shenzhen have been unified into one standard price. The flag-fall prices were CNY 10 per 2 km and CNY 2.60 per km for over 2 kms. The location of the study area is shown in Figure 1.

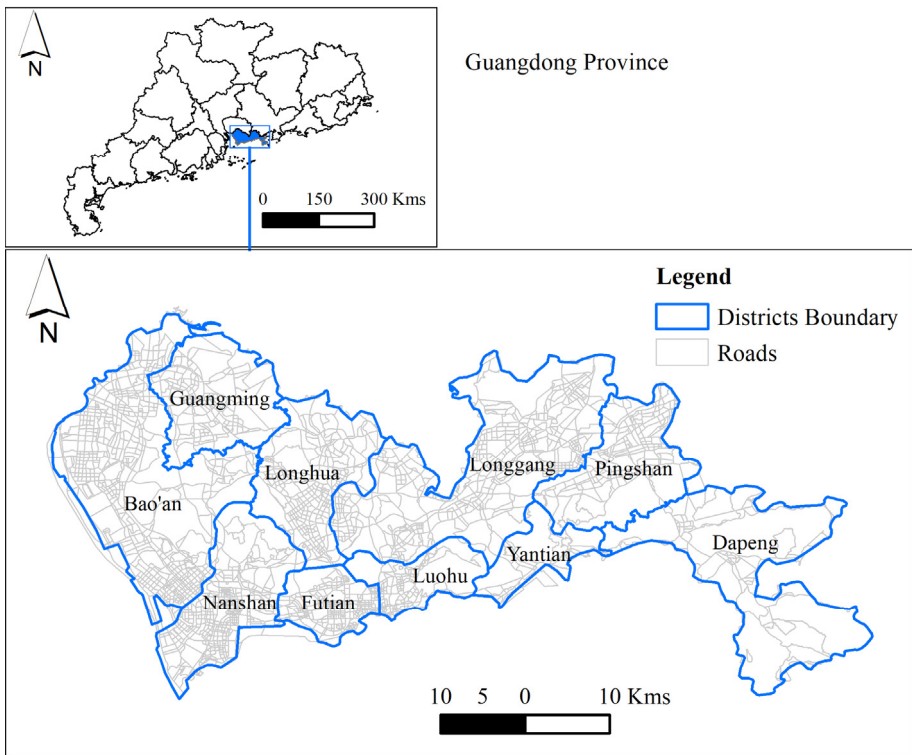

**Figure 1.** The study area map (the geographical background is obtained by vectorization from Shenzhen map provided by Shenzhen Municipal Bureau of planning and natural resources).

In accordance with Shenzhen transport annual report 2016 [50], the average daily travel of Shenzhen residents reached 44.43 million trips in 2016, including 21.33 million motorized trips. Shenzhen's motorized travel modes include conventional buses, customized buses, rail transit, cruised taxis, online car-hailing, private cars, institutional shuttle buses, etc. In 2016, the number of motor vehicles in Shenzhen was 3,225,800. The modal split showed that 28.0% of the residents traveled by conventional buses, 0.3% by customized buses, 13.6% by rail transit, 3.7% by cruised taxis, 4.3% by online car-hailing, 41.1% by private cars, 6.6% by institutional shuttle buses, and 2.5% by other modes. The proportion of motorized travel by public transportation, including conventional buses, customized buses, rail transit, cruised taxis, online car-hailing, and institutional shuttle buses is 56.5%, which maintains the dominant position in the city's modal split. In 2016, the average speeds during the morning and evening peak hours of weekdays were 30.6 and 26.0 km/h, respectively. During the evening peak hours on weekdays, the average speeds in Luohu District, Longgang District, and Longhua District were 23.7, 25.5, and 25.6 km/h, respectively. The average speeds in Dapeng New District, Pingshan District, and Guangming District were 42.6, 38.0, and 31.8 km/h, respectively.

### 3.2. Data Source and Preprocessing

The taxi trip data of Shenzhen city were provided by major taxi operation companies, which cover almost 65% of the total taxicabs in Shenzhen. The data involved in our study were collected in 10, 11, 15, and 16 April 2017, and the former two days were weekdays, while the last two days were weekends. The fields of the taxi data are vehicle ID (anonymized), pick-up time, drop-off time, latitude and longitude of pick-up and drop-off points, trip date, trip distance, and trip duration. The coordinate system of the longitude and latitude fields is GCJ-02, which is the officially Chinese geodetic datum formulated by the Chinese State Bureau of Surveying and Mapping. The coordinates of pick-up and drop-off points were converted to WGS 1984 coordinate system. Since our study focuses on the taxi travel characteristics within Shenzhen city, all non-compliant trip records were excluded.

The original POI data were collected from Gaode Map (https://lbs.amap.com/, accessed on 7 December 2020) in 2020, which includes fourteen POI categories: Catering facilities, scenic spots, public service facilities, companies, shopping facilities, transportation facilities, financial facilities, educational, scientific and cultural facilities, residence district, living service facilities, sports and leisure facilities, medical service facilities, government agencies, and accommodation service facilities. First, the GCJ-02 coordinates of POIs were converted to WGS 1984 coordinates. Second, duplicate POIs and POIs with low public recognition, such as public toilets, newsstands, etc., were eliminated. Third, the POIs were reclassified in accordance with the latest urban land classification and planning and construction land standards [51]. To facilitate further analysis, the abbreviations of urban land types were used. For example, public administration and public service land are abbreviated as public service area, and green space and square land are abbreviated as greenspace area. The POIs were reclassified into six categories of land use: Traffic service area, public service area, commercial service area, residential area, industrial area, and greenspace area. The ultimate classification results of POIs are as follows: 37,541 records of traffic service area POIs, 68,589 records of public service area POIs, 163,529 records of commercial service area POIs, 16,016 records of residential area POIs, 146,826 records of industrial area POIs, and 2502 records of greenspace area POIs.

The land use data used in our study were the result of the Third National Land Resources Survey of China, with a spatial resolution of 10 m and the survey year was 2017 [52]. The land use types include residential land, commercial service land, industrial land, transportation land, public administration land, grassland, etc.

The road network used in this paper was the OSM road network data, which was collected in 12 July 2020 (https://www.openstreetmap.org/, accessed on 12 July 2020). First, the coordinate system of road network data was converted to WGS 1984 coordinate

system. Then, short and low-grade roads were removed. Next, tools of ArcGIS software (Esri, Redlands, CA, USA), such as buffer, dissolve, and collapse-dual-lines-to-centerline were used to remove duplicate roads. Finally, topology errors were corrected to obtain the ultimate road network data.

Traffic analysis unit is the geography unit, which is most commonly used in transportation analysis. The type and spatial extent of zones usually varies in different researches. The analysis unit derived by road network is one of the classic geography units for transportation analysis. In addition, the unit created by road network is also suitable for UFA research. Therefore, the unit created by the road network was selected as the analysis unit of this study. First, the preprocessed OSM road network was used to segment the administration boundary of Shenzhen. Then, we used the tools of ArcGIS software (Esri, Redlands), such as line to polygon, eliminate, and dissolve to generate the analysis units for UFA identification.

### 3.3. Methods

The methodological steps of our study are shown in Figure 2.

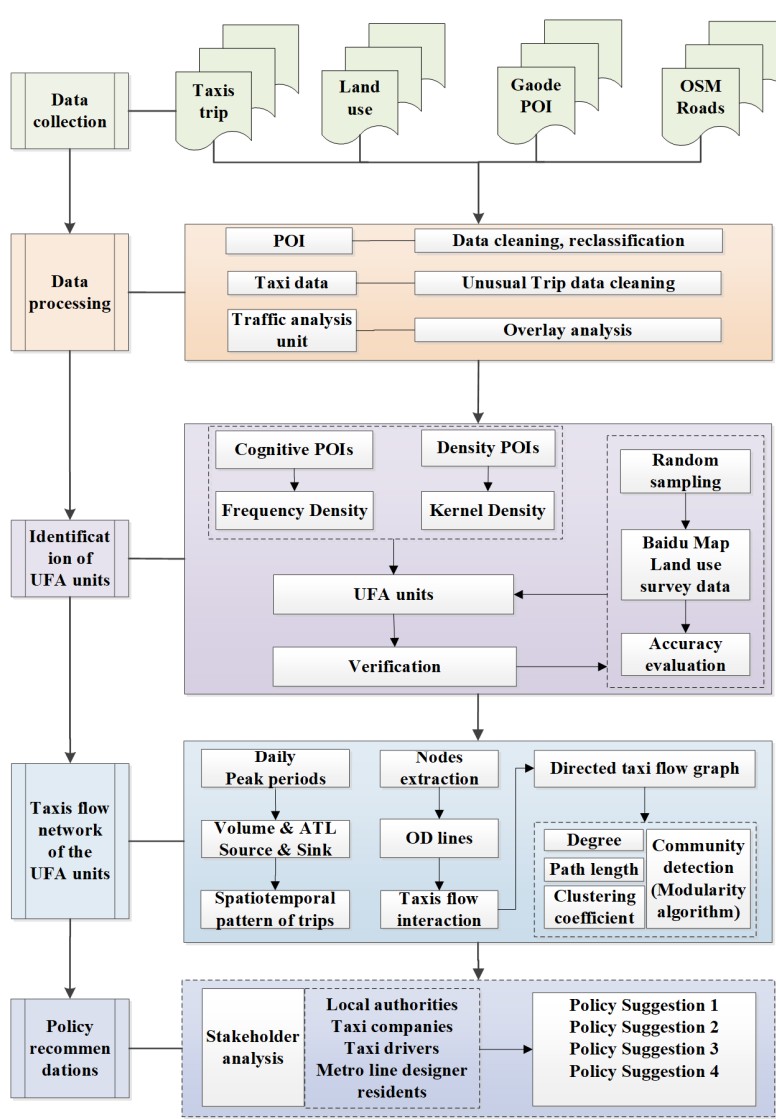

**Figure 2.** The data flow chart of the study.

### 3.3.1. Method for Urban Functional Area Identification

In accordance with the public recognition, footprint, and distribution patterns of geographic entities, we divided POIs into two major types (cognitive POIs and density POIs) and applied different identification methods, respectively. The cognitive POIs represent the geographic entities, which usually have a relatively large area and high social recognition. Specifically, these POIs can usually dominate the UFA units where they are located, such as railway stations, grade A tertiary hospitals, parks, etc. Therefore, the frequency density and type ratio formula were used for these POIs. The density POIs refer to geographic entities with relatively small area and low social recognition, such as convenience stores, bistro, barber store, etc. Due to the large number of these POIs and their scattered distribution, kernel density estimation method was applied to determine the dominant function of the UFA unit. The identification methods for two major types of POIs are shown in Table 1.

**Table 1.** The identification methods for two major types of POIs.

| Major Type | POI Categories | Methods |
|---|---|---|
| Cognitive POIs | Train Station, Bus Stops, Government Organization, Grade A tertiary hospital, Universities and Colleges, Parks and Piazzas, Scenic Spots, etc. | Frequency Density (FD) Category Ratio (CR) |
| Density POIs | Catering facilities, Shopping facilities, Financial facilities, Living Service facilities, Sports facilities, Transportation facilities, Public facilities, Scientific, educational and cultural institutions, Medical facilities, Companies, Residential districts, etc. | Kernel Density |

- Identification method for Cognitive POI-dominated UFA units

In accordance with previous studies [53,54], different weights were set for each type of POIs based on the following principles: Higher weights for POIs with small number and high social recognition in the city, while lower weights for POIs with large number and low social recognition. The specific weights setting was 20 for commercial service land, 30 for residential land, 50 for public service land, 75 for industrial land, 90 for greenspace and scenic land, and 100 for transportation service land.

For cognitive POI-dominated UFA identification, the frequency density (FD) and category ratio (CR) formulas were used to identify the nature of unit. The calculation formulas are shown as follows:

$$F_i = \frac{n_i}{Ni}(i = 1,\ 2,\ 3\ldots,\ 6), \tag{1}$$

$$C_i = \frac{F_i}{\sum_{i=1}^{6} F_i} \times 100\%(i = 1,\ 2,\ 3\ldots,\ 6) \tag{2}$$

In Formula (1), $i$ denotes the type of POI (commercial, transportation, public, industrial, residential, and greenspace); $n_i$ denotes the number of the $i$-th type of POI in the UFA unit; $Ni$ denotes the total number of the $i$-th type of POI in the study area; $F_i$ denotes the frequency density of the $i$-th type of POI in the UFA unit to the total number of the $i$-th type of POI in the study area, and FD for short. In Formula (2), $C_i$ denotes the ratio of the frequency density of the $i$-th type of POI to the frequency density of all POIs in the UFA unit, and CR for short.

The number of each POI type in each UFA unit was calculated using the Spatial Join tool of ArcGIS software (ESRI, Redlands), and the FD and CR of each POI type in each UFA unit were calculated using the Field Calculator tool of ArcGIS software (ESRI, Redlands). In accordance with previous studies [20] and our experiment results, the rule was set that when the CR value of cognitive POIs is greater than 30%, the UFA unit is identified as a single type UFA unit that is dominated by one functional type.

- Identification method for Density POI-dominated UFA units

The density POI-dominated UFA unit was identified using the kernel density estimation (KDE) method, which is widely used in the study of geography. The kernel density

analysis method is derived from the first law of geography, which states that things are more closely connected the closer they are, and the closer location to the core element acquires a greater value of density expansion. The formula of kernel density is shown as follows:

$$f(x) = \sum_{i=1}^{n} \frac{1}{h^2} K\left(\frac{x - x_i}{h}\right), \tag{3}$$

$$K\left(\frac{x - x_i}{h}\right) = \frac{3}{4}\left(1 - \frac{(x - x_i)^2}{h^2}\right) \tag{4}$$

In Formulas (3) and (4), $K$ is the kernel function; $h$ is the search radius or bandwidth, which determines the smoothness of the surface of the kernel density; and $n$ is the number of elemental points contained in the search radius at point $x$.

Previous studies demonstrated that the bandwidth has a large effect on the results of kernel density estimation. Specifically, Ding et al. claimed that a bandwidth in the range of 200–300 m is appropriate when performing kernel density analysis [55]. Therefore, the bandwidth was set to 200 m in this study, the output image element size was the default size, and the natural breakpoint method was used for classification. Since the analysis results are greatly influenced by the number of POIs, the weights of different POI types were adjusted in accordance with the characteristics of each POI type. The specific weights were set as follows: 70 for greenspace facilities, 50 for public service facilities, 15 for commercial service facilities, 35 for transportation facilities, 30 for industrial facilities, and 100 for residential districts. The kernel density estimation results were normalized using the Fuzzy Membership tool to ensure that the density values fall into the [0, 1] interval.

Finally, the Field Calculator tool was used to calculate the total density of each UFA unit. In accordance with previous studies, 50% was used as the threshold value [20]. Specifically, when the POI density value of one type of function within the analysis unit exceeded 50% of the total density, the unit was identified as a single-type UFA unit. In contrast, the unit was identified as a mixed-type UFA unit, which is dominated by the top two urban functional types in density values.

### 3.3.2. Source-Sink Analysis Method

To understand whether different types of UFA units are inflow or outflow areas for taxi travel, the source-sink analysis was performed. As known, the volume of taxi trips shows a more pronounced temporal divergence in the 24-h period. Particularly in the morning and evening peak hours, the volume of taxi flow is highest and most significantly different from other time periods. In accordance with the morning and evening peak restriction policies in Shenzhen [56], the morning peak hours are from 07:00 a.m. to 09:00 a.m., and the evening peak hours are from 17:30 p.m. to 19:30 p.m. Therefore, this study extracts the morning and evening peak hours data to study the travel source-sink characteristics at the scale of UFA units. In reference to the study by Gao et al. [57], each UFA unit was divided into three categories in accordance with its net inflow and net outflow values: Source area (net outflow rate > 0), sink area (net inflow rate > 0), and equilibrium area (net inflow rate and net outflow rate are zero). The net inflow rate or net outflow rate [58] was calculated as follows:

$$P_i = \frac{(D_i - O_i)}{(D_i + O_i)}, \tag{5}$$

In Equation (5), $P_i$ is the net inflow (outflow) rate of the $i$-th functional area unit in the study period; $D_i$ and $O_i$ represent the total number of taxis inflowing and outflowing in the functional area unit, respectively. The closer the $P_i$ to 1, the more significant the inflow of taxis in the unit, and the closer the $P_i$ to $-1$, the more significant the outflow of taxis in the analysis unit.

### 3.3.3. Complex Network Analysis Method

In our study, the software for complex network analysis was Gephi 0.9.2, which is a popular opensource software (https://gephi.org/, accessed on 21 December 2021). Each UFA unit obtained in the previous section was used as a node to calculate the OD matrix of taxi trips and construct a complex network. The metrics, including degree distribution, average path length, and clustering coefficient, were used in our study to explore the taxis flow network. In addition, the network communities were detected using the modularity algorithm.

The degree of a node in a network is the number of other nodes, which is directly connected to the node. The degree of a node is positively correlated with the importance of the node. The higher the degree of a node, the more important it is in the network. In a directed network, the degree of a node includes out-degree and in-degree. The degree of a node is defined by the following equation:

$$D_i = \sum_{j=1}^{n} w_{ij}, \tag{6}$$

where $D_i$ is the degree of node *I*; $n$ is the total number of nodes; and $w_{ij}$ denotes the weight of the connected edges between node $i$ and node $j$.

The average path length of a network ($L$) is the average distance between any two nodes, which reflects the degree of clustering and dispersion of the network. $L$ was calculated as follows:

$$L = \frac{1}{n(n-1)} \sum_{ij}^{n} d_{ij}, \tag{7}$$

where $d_{ij}$ is the shortest path between node $i$ and node $j$; and $n$ is the total number of nodes in the network.

The clustering coefficient ($C$) is a measure of the average probability that two neighboring nodes of a node are also neighbors of each other, in effect measuring the density of the triangular structure in the network [10]. The clustering coefficient can be used to describe the compactness of the network. The formula of clustering coefficient is shown as follows:

$$C = \frac{1}{n} \sum_{i=1}^{n} C_i, \tag{8}$$

$$C_i = E_i / E_{k_i}^2, \tag{9}$$

where $C_i$ is the clustering coefficient of node *I*; $C$ denotes the clustering coefficient of the network, which is the average of the clustering coefficients of all nodes in the network; $E_i$ is the number of node pairs directly connected in the neighboring nodes of node *I*; and $E_{k_i}^2$ denotes the total number of neighbor node pairs of node $i$.

In a network, a subgraph consisting of nodes with similar attributes and connected edges between nodes is called a community. The community structure of taxi flow network was detected using the modularity algorithm proposed by M.E.J. Newman [59]. The modularity can be calculated as follows:

$$Q = \frac{1}{2m} \sum_{ij} \left[ A_{ij} - \frac{k_i k_j}{2m} \right] \delta(C_i, C_j), \tag{10}$$

In Formula (10), $Q$ is the modularity; $m$ is the number of edges in the network; $A_{ij}$ denotes the element of network adjacency matrix; $k_i$ and $k_j$ represent the degree values of node *I* and node $j$; $C_i$ is the community that node $i$ belongs to; and $\delta(C_i, C_j)$ is a function, which indicates whether two communities are the same (if $C_i = C_j$, $\delta(C_i, C_j) = 1$, otherwise $(C_i, C_j) = 0$). The value of modularity ranges from 0 to 1. A higher value of modularity indicates a more reasonable community division result.

## 4. Results

### 4.1. Identification Result of UFA Units

As described in Section 3, the identification results of cognitive POIs and density POIs were combined to produce the ultimate urban functional area units. The results showed that 2565 UFA units were identified, including 21 UFA types. Among them, the number of single type UFA units was 880 and the number of mixed type UFA units was 1685. Each UFA type was given an abbreviation, respectively (see Table 2). For example, the PRFA represented a mixed type UFA unit dominated by public service function and residential function, the IRFA represented a mixed type UFA unit dominated by industrial function and residential function. One hundred UFA units were randomly selected, and the accuracy of the identification results was evaluated using the land use survey data as the true value. The overall accuracy of the identification results reached 75%, indicating that our method can effectively and accurately identify the UFA units in Shenzhen. The statistics of identification results are shown in Table 2.

**Table 2.** The statistics of identification results.

| UFA Type | Abbreviation | Amount | Total Area (km$^2$) | Average Area (km$^2$) |
|---|---|---|---|---|
| public service and residential functional area | PRFA | 544 | 242.65 | 0.45 |
| industrial functional area | IFA | 248 | 140.05 | 0.56 |
| public service and commercial functional area | PCFA | 237 | 82.27 | 0.35 |
| public service and industrial functional area | PIFA | 232 | 130.37 | 0.56 |
| industrial and residential functional area | IRFA | 229 | 184.37 | 0.81 |
| greenspace and scenic spot functional area | GSFA | 202 | 864.5 | 4.28 |
| residential functional area | RFA | 194 | 55.16 | 0.28 |
| public service functional area | PFA | 139 | 64.76 | 0.47 |
| commercial service and industrial functional area | CIFA | 130 | 73.94 | 0.57 |
| residential and commercial service functional area | RCFA | 121 | 41.37 | 0.34 |
| transportation and industrial functional area | TIFA | 59 | 49.64 | 0.84 |
| transportation and public service functional area | TPFA | 56 | 27.12 | 0.48 |
| commercial service functional area | CFA | 51 | 6.51 | 0.13 |
| transportation service functional area | TFA | 46 | 14.73 | 0.32 |
| residential and transportation service functional area | RTFA | 39 | 6.14 | 0.16 |
| commercial service and transportation functional area | CTFA | 15 | 1.64 | 0.11 |
| public service and greenspace functional area | PGFA | 8 | 2.38 | 0.3 |
| industrial and greenspace functional area | CGFA | 6 | 2.73 | 0.46 |
| greenspace and transportation functional area | GTFA | 4 | 2.67 | 0.67 |
| greenspace and residential functional area | GRFA | 4 | 1.4 | 0.35 |
| greenspace and commercial functional area | GCFA | 1 | 0.16 | 0.16 |

In Table 2, it can be seen that the number of mixed type UFA units in Shenzhen was about twice the number of single type UFA units, indicating that the city is mainly dominated by mixed type UFA units. This result also reflected the tendency of mixed and intensive land use in the development process of Shenzhen. The number of UFA units mixed with residential functions was the largest, which include PRFA, IRFA, RCFA, and RTFA, with numbers 544, 229, and 121, respectively. We believe that the result is reasonable as residence is one of the basic needs of human life. The number of IFA was 248, ranking 2nd among all UFA types. In addition, the UFA units mixed with industrial function mainly include PIFA, IRFA, CIFA, and TIFA, whose numbers were 232, 229, 130, and 59, respectively. The high proportion of industrial related UFA units indicated, to a certain extent, that industrial land occupied an important position in the land use structure of Shenzhen. UFA units only providing commercial service were rare and usually appeared in the mixed type UFA units. The commercial service function was mainly mixed with public service, industrial function, and residential service, and the numbers of mixed type UFA units were 237, 130, and 121, respectively. UFA units only providing transportation service were also relatively small, and mainly mixed with industrial, public service, and residential functions. The number of UFA units with greenspace and scenic function was

202, and rarely mixed with other functions. In terms of average area, the average area of GFA was significantly higher than the other UFA types, and was followed by TIFA, IRFA, GIFA, and CTFA. The distribution of UFA units is shown in Figure 3.

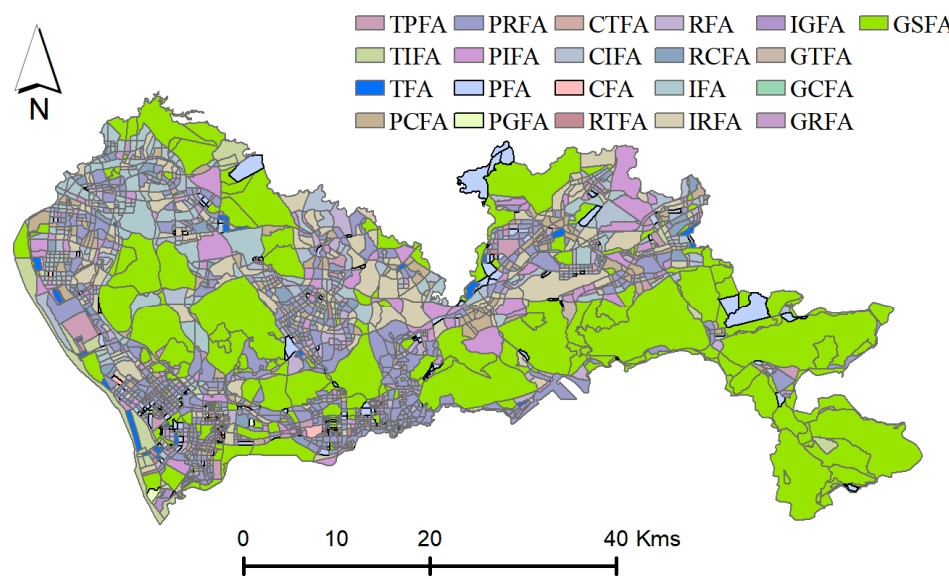

**Figure 3.** The identification results of urban functional area (the geographical background is obtained by overlaying roads and administration boundary of Shenzhen city).

In Figure 3, it can be seen that the single type UFA units were mainly located in the central and eastern parts of Shenzhen, which is the ecological protection zone of Shenzhen and the terrain is mainly mountainous and hilly. The mixed type UFA units were mainly located in the northern, western, and southern parts of Shenzhen, where the topography is mainly plain, the built-up area is large, and the economy is more developed.

### 4.2. Spatiotemporal Characteristics of Taxi Trips at the Scale of UFA Units

The daily travel volume and average trip length (ATL) in weekdays and weekends are shown in Figures 4 and 5 (distinguished pick-up and drop-off), respectively.

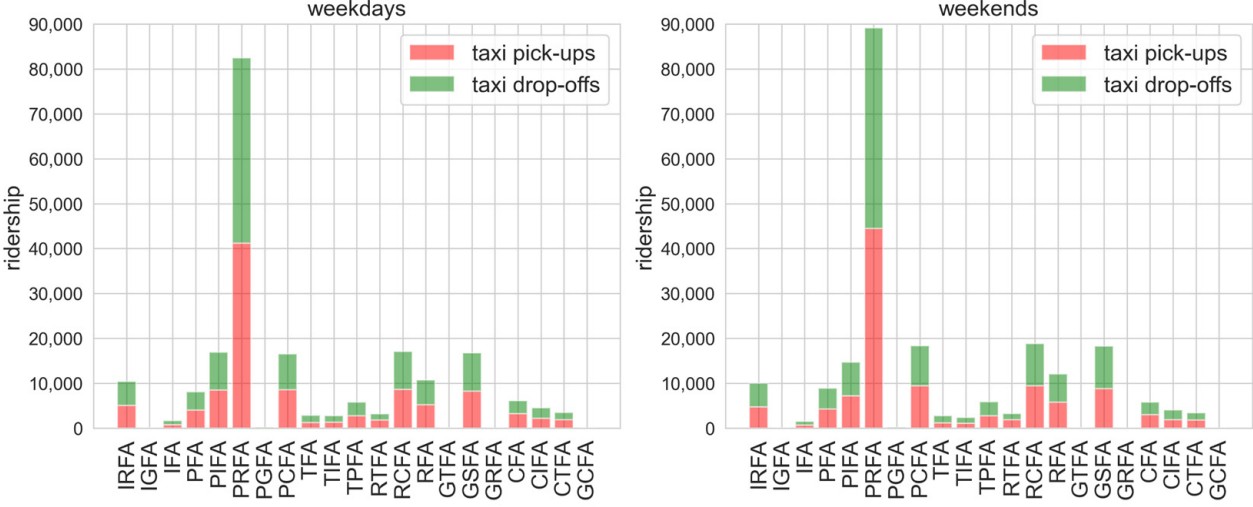

**Figure 4.** Daily average travel volume of each UFA type on weekdays and weekends.

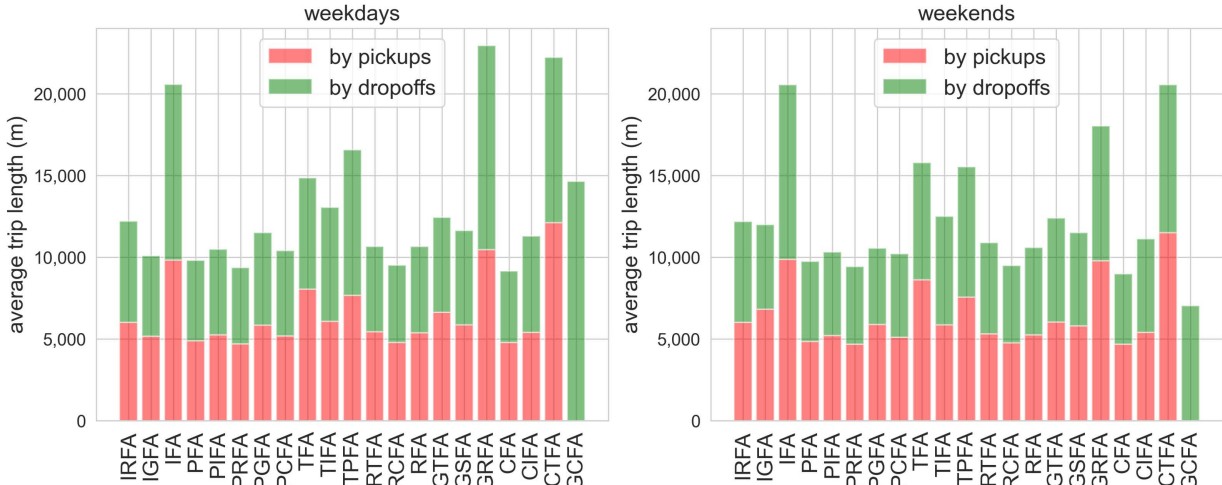

**Figure 5.** Daily average travel distance of each UFA type on weekdays and weekends.

In Figure 4, it can be seen that the average count of pick-ups and drop-offs in the UFA units, including PRFA, PCFA, RCFA, and GFA, were higher during weekends than during weekdays, while the average ridership in the PIFA was contrary. This result indicated that taxi travel behavior was more active in UFA units dominated by residential, commercial, and scenic greenspace functions during weekends, while more active in UFA units dominated by industrial function during weekdays. The reason may be that commuting behavior dominates during weekdays, thus trips were more active in mixed type UFA units dominated by industrial function during weekdays. As people's leisure and recreational activities increased during weekends, taxi trips in mixed type UFA units, including commercial and scenic functions increased significantly.

In Figure 5, it can be seen that whether during weekdays or weekends, the average travel distance in the UFA units, including IFA, TFA, TPFA, GRFA, and CTFA, were relatively long, while relatively short in the UFA units, such as PRFA and RCFA. In terms of travel distance corresponding with pick-up points, the average travel distance of various UFA types during weekends and weekdays was relatively close. The average travel distance of UFA units, such as IFA and TFA was relatively long during weekends, while the average travel distance of UFA units, such as GTFA, GRFA, and CTFA was relatively long during weekdays. In terms of the travel distance corresponding with the drop-off point, the average travel distance in the UFA units, including PGFA, TGFA, GRFA and GCFA, was significantly longer during weekdays than during weekends.

The map of taxis mobility index during the morning and evening peak periods is shown in Figure 6 (the UFA units in light gray were missing data).

In Figure 6, it can be seen that during the morning peak period, the UFA units with higher taxi mobility were mainly concentrated in the central and western part of Shenzhen, as well as some units in the northern part. Specifically, the sink UFA units were mainly distributed in Futian District and Nanshan District. The peripheral areas of Futian District and the residential function related UFA units in Luohu District adjacent to Futian District were the main sources of the flow of taxis. The mobility index of Nanshan District showed an overall pattern of "the central region was the source area, the surrounding region was the sink area". The source and sink areas in Baoan District showed a staggered distribution, and the convergence phenomenon near Shenzhen Baoan international airport was significant. The southwestern and northeastern part of Longgang District, which is dominated by residential function, was the main source area during the morning peak period.

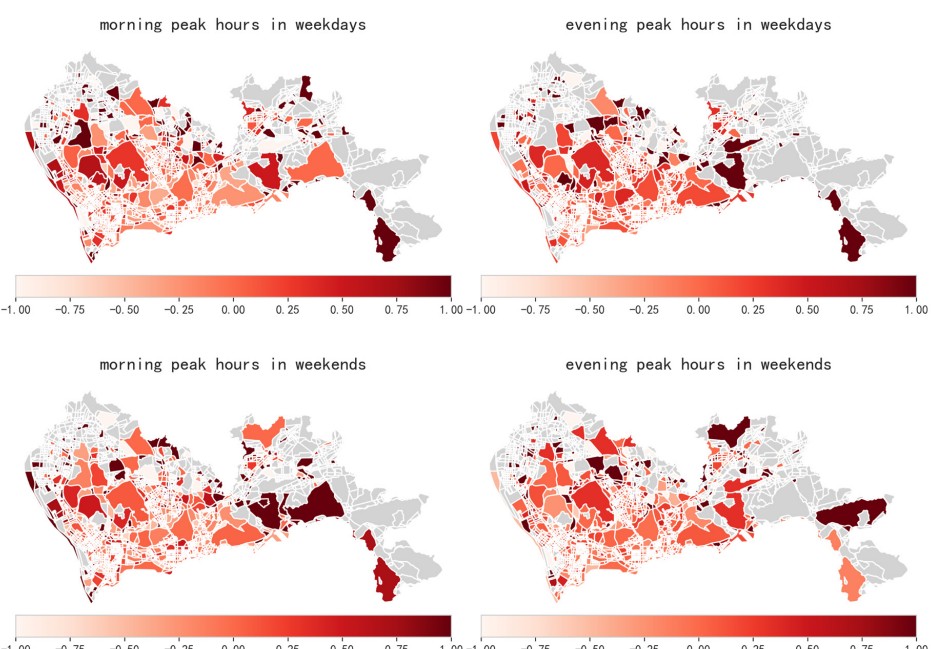

**Figure 6.** The map of taxis mobility index during the peak periods (the geographical background is obtained by overlaying roads and administration boundary of Shenzhen city).

Contrary to the morning peak period, the central part of Futian District and Nanshan District turned into the main source areas, and the sink areas were distributed around the above-mentioned areas. The sink areas were mainly concentrated in the southern part of Yantian District, the southwestern part of Longgang District, and the eastern part of Luohu District. The reason may be that these areas are mainly dominated by residential mixed land use types. The source sink area also showed a disorderly pattern of staggered distribution in the southern part of Baoan District.

*4.3. Characteristics of Taxi Mobility Network among the UFA Units*

The chord grams of traffic flow between different UFA types on weekdays and weekends are shown in Figure 7.

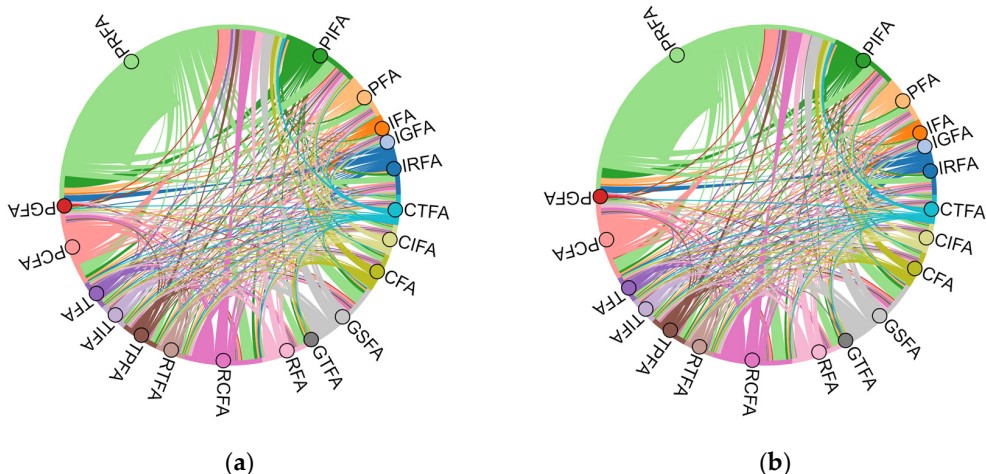

(**a**)                       (**b**)

**Figure 7.** The taxi flow interaction between different UFA types: (**a**) Chord gram on weekdays; (**b**) chord gram on weekends.

In Figure 7, it can be seen that the interaction characteristics between the different UFA unit types during weekdays were very similar with the interaction characteristics during weekends. The top three UFA types in terms of outflow were IRFA, RCFA, and

PCFA, with outflow values of 36,548, 7668, and 7538, respectively. The bottom three UFA types in terms of outflow were GRFA, IGFA, and GTFA, with outflow values of 3, 15, and 26, respectively. The top three destination UFA types outflowing from the IRFA were IRFA, RCFA, and PCFA, with outflow values of 15,414, 2903, and 2807, respectively. The top three destination UFA types with outflows from the RCFA were PRFA, RCFA, and PIFA, with outflow values of 2986, 716, and 638, respectively. The top three destination UFA types with outflows from the PCFA were PRFA, PCFA, and RCFA, with outflow values of 2888, 766, and 622, respectively.

The top three UFA types in terms of inflow were PRFA, GSFA, and PIFA, with outflow values of 36,081, 7660, and 7628, respectively. The top three UFA types with PRFA as the destination of inflow were PRFA, RCFA, and PCFA, with inflow values of 15,414, 2986, and 2888, respectively. The top three UFA types with GSFA as the inflow destination were PRFA, GSFA, and PIFA, with inflow values of 2793, 864, and 692, respectively. The top three UFA types with PIFA as the inflow destination were PRFA, PIFA, and GSFA, with inflow values of 2793, 864, and 692, respectively.

The daily network flow diagram between each UFA unit is shown in Figure 8.

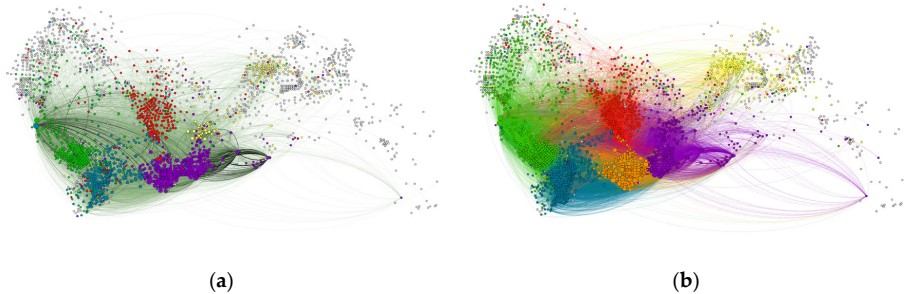

(**a**)                                    (**b**)

**Figure 8.** The directed network of daily taxi flow: (**a**) Network on weekdays; (**b**) network on weekends.

In Figure 8, it can be seen that the daily traffic flow network during weekdays showed a spatial pattern of "dense in the west and south, sparse in the east and north", indicating that the taxi flow of Shenzhen residents showed significant spatial imbalance. The nodes of network showed a pattern of "polycentric and concentrated distribution in central areas". The network was in a converged state, and the spatial characteristics showed an intermittent polar nucleus shape, i.e., grouped distribution. The quantity of interactions in a particular area was particularly high, and the nodes varied significantly in size. The nodes in the regions, such as downtown business service area, industrial park, and important transportation service area were the hot spots of city during weekdays.

Compared with weekdays, the daily travel network during weekends was more dispersed, with spatial characteristics of continuous spread. Meanwhile, the differences in node sizes decreased, indicating that residents' travel destinations were more diverse. The results showed a decrease in the amount of taxi interactions between different UFA units on weekends. The UFA units, such as public service and greenspace, scenic areas, and residential areas turned into hotspots of the city during weekends. Compared with weekdays, the edge color in Yantian District was significantly darker, indicating an increase in the quantity of taxi interactions. The reason may be that Yantian District has many tourist attractions represented by the Dameisha seaside park, thus many residents visit Yantian District for recreation during weekends. Another phenomenon was that the values of taxi inflow and outflow at important transportation nodes, such as Shenzhen Baoan international airport and Shenzhen North railway station decreased significantly during weekends. The reason may be that during weekdays these transportation sites were important nodes connecting residents of nearby cities, such as Guangzhou, Shenzhen, and Dongguan, and residents who commute across the city tend to change cabs at these sites. However, the intermediary role of the airport and the railway station diminished substantially during weekends.

In terms of the node pattern, the nodes in Futian District and Luohu District were larger in size and more densely distributed, indicating more frequent taxi movements within these two districts. Specifically, there were many popular locations in the two districts, such as Huaqiang North shopping district, Fairy Lake botanical garden, Futian port of entry, and Shenzhen railway station, which were typical nodes in the network. The distribution of nodes within the Baoan District, Nanshan District, and Longhua District were more concentrated but smaller in size, indicating that the volume of taxi travel in these areas was relatively low. The nodes in the remaining areas were more scattered and smaller in size. In addition, numerous taxi trips interacted between two important transportation service nodes, including Shenzhen BaoAn international airport and Shenzhen North railway station and occurred in the regions, including Nanshan District, Futian District, and Luohu District.

The network flow diagram between each UFA unit during the peak periods of weekdays are shown in Figure 9.

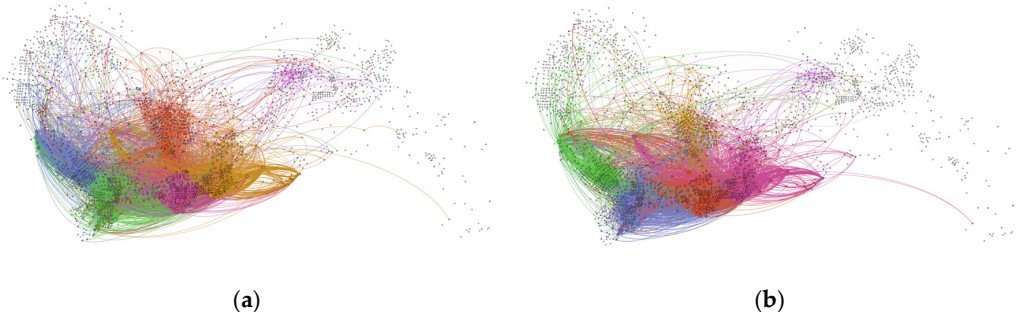

(**a**)　　　　　　　　　　　　　　　(**b**)

**Figure 9.** The directed network of taxi flow in peak periods of weekdays: (**a**) Network in morning peak hours of weekdays; (**b**) network in evening peak hours of weekdays.

In Figure 9, during the weekday morning peak, the node sizes were larger in areas, such as the central and southern part of Futian District and the southern part of Luohu District, while the node sizes in the remaining areas were smaller. This result indicated that the hotspot areas during the morning peak were mainly distributed in Futian District and Luohu District, which have the most prominent commercial and trade service functions in Shenzhen. Due to the strong comprehensive economic strength of Futian District and Luohu District and their geographical location near the port, taxi interactions within and between the two districts were more frequent. Compared with the morning peak, the node size in the southern part of Futian District was smaller during the evening peak, indicating a decrease in taxi travel volume in the region. Meanwhile, the node size in the southern part of Luohu District was increased, indicating an increase in taxi travel in the region. The color depth of the edges showed that the taxi flow between Futian District and Luohu District increased during the evening peak, especially for long distance trips.

The network flow diagram between each UFA unit during peak periods of weekends are shown in Figure 10.

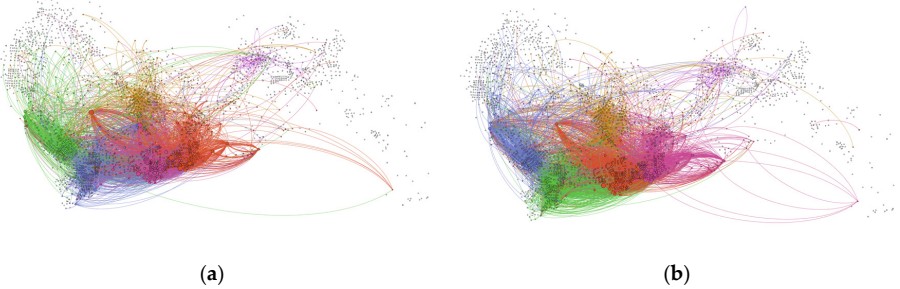

(**a**)　　　　　　　　　　　　　　　(**b**)

**Figure 10.** The directed network of taxi flow in peak periods of weekends: (**a**) Network in morning peak hours of weekends; (**b**) network in evening peak hours of weekends.

Compared with the morning peak period on weekdays (see Figure 9), it can be seen in Figure 10 that the node degree values dropped significantly during the morning peak period of weekends, and the edge distribution was sparser. In particular, the node sizes in the southern and central part of Futian District were significantly decreased, while the distribution was sparser, indicating a significant decrease in taxi traffic in the central and southern part of Futian District. Compared with the same period on weekdays, the sizes of two nodes, including Shenzhen North railway station and Shenzhen railway station, were larger. The reason may be due to the large number of urban migratory birds in Shenzhen. Due to the high housing price in Shenzhen, people choose to buy houses in neighboring cities (e.g., Dongguan, Huizhou, etc.). These people live in Shenzhen during weekdays and return to neighboring cities by intercity trains during weekends, thus causing a huge taxi flow at traffic nodes, such as Shenzhen North railway station during the weekend morning peak. During the weekend evening peak, there were fewer extra-large nodes in the network, mainly distributed in the central part of Futian District, central and southern part of Luohu District, with smaller and more evenly distributed node sizes in the remaining areas. Compared with the weekend morning peak period, the distribution of edges in the network was denser, indicating more taxi interactions, especially that the interaction between Yantian District and Luohu District was significantly increased.

The statistical information of networks is shown in Table 3.

**Table 3.** Statistical description of taxi flow network.

| Period | Average Degree | Average Path Length | Clustering Coefficient |
| --- | --- | --- | --- |
| Weekdays daily | 24.016 | 2.771 | 0.267 |
| Weekends daily | 28.396 | 2.711 | 0.294 |
| Morning peak on weekdays | 3.863 | 3.554 | 0.103 |
| Evening peak on weekdays | 3.406 | 3.590 | 0.118 |
| Morning peak on weekends | 2.716 | 3.746 | 0.094 |
| Evening peak on weekends | 4.134 | 3.471 | 0.134 |

In Table 3, it can be seen that the average path length of weekends was smaller than weekdays, while the clustering coefficient was larger than weekdays. This result indicated that the taxi mobility network on weekends was more consistent with the characteristics of a small world network than during weekdays. The average path length on weekdays was slightly larger than the average path length on weekends, indicating that the network connectivity diversity was stronger on weekdays, with longer interaction distances and wider connectivity between different UFA units. The average degree of weekends was greater than weekdays, indicating that the average number of connected edges of the taxi mobility network was greater during weekends than during weekdays, and the network was more stable than weekdays.

In accordance with the metrics of network in morning and evening peaks of weekdays, we found that the average degree and the average number of connected edges were higher during the weekday morning peak than during the weekday evening peak, indicating that the influence of commuting behavior was more frequent during the morning peak. The average path length and clustering coefficients of the network during the evening peak were larger than those during the morning peak, indicating that the average interaction distance between UFA nodes was longer during the evening peak and the network was denser during the evening peak.

By comparing the characteristics of network during the weekend peak periods, we found that the average degree value of the evening peak was significantly larger than the morning peak, indicating that residents were more active in traveling during the weekend evening peak. The average path length during the weekend morning peak was larger than the evening peak, while the clustering coefficient was smaller than the morning peak. For this reason, it can be concluded that the network diameter during the weekend morning peak was larger, and the interaction distance was longer. Moreover, the network was

sparser compared with the evening peak, while the small world characteristics of the network during the evening peak were more significant.

The community detection results of taxi flow network are shown in Figure 11. The modularity values of community detection results on weekdays and weekends were 0.426 and 0.435, respectively. Due to the fact that modularity values of two networks were both larger than 0.4, we claimed that the results of community detection are reasonable. The number of communities on weekends was less than weekdays, which can, to a certain extent, indicate that the travel range of residents was more concentrated during weekends.

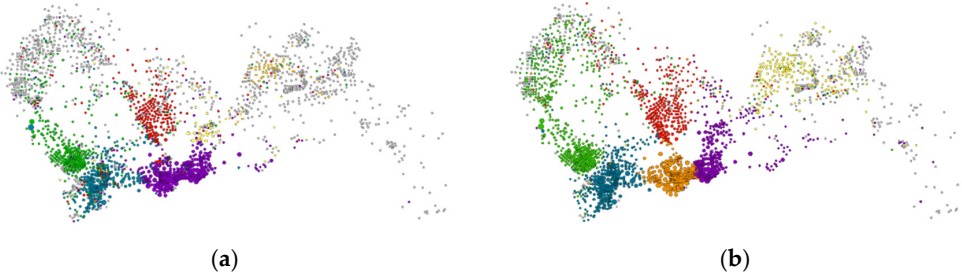

(**a**)         (**b**)

**Figure 11.** Community detection results of taxi daily flow network: (**a**) Community of weekdays; (**b**) community of weekends.

In Figure 11, it can be seen that the results of community detection on weekdays and weekends showed different characteristics, and the differences in community morphology and the number of nodes reflected the spatiotemporal heterogeneity of the network structure. The taxi mobility network during weekdays was divided into five neighborhoods, and the distribution of neighborhoods was more consistent with the administrative division. This result indicated that taxi trips during the weekday period were predominantly within the districts. Compared with weekdays, the community characteristics during weekends showed significant differences. Futian District and Luohu District were divided into two communities from one community during weekdays, and Baoan District and Longgang District have expanded community boundaries, indicating more frequent taxi movements within the administrative districts during weekends. In addition, Luohu District and the southern part of Longgang District formed one community with higher intra-regional mobility than during weekdays.

## 5. Conclusions and Prospect

This paper analyzed the spatiotemporal characteristics of taxi trips in Shenzhen, China by applying the complex network analysis method in the scale of urban functional area unit. The main findings can be summarized as follows:

(1) Taxi travel behavior is more active in urban functional area units dominated by residential, commercial, scenic, and greenspace functions during weekends, and more active in urban functional area units dominated by industrial function during weekdays. In the mixed type urban functional area units dominated by residential function, the travel volume is slightly higher on weekends than the travel volume on weekdays. In the mixed type urban functional area units dominated by scenic and greenspace function, the travel volume on weekends is significantly larger than the travel volume on weekdays. The urban functional area units of public service are one of the main travel destinations, and taxi travel in these units are more active during weekends than weekdays.

(2) During the morning peak period, the urban functional area units with higher taxi mobility are mainly concentrated in the central and western part of Shenzhen, as well as some units in the north. The urban functional area units as sink of flow are mainly distributed in Futian District and Nanshan District. During the evening peak period, the sink areas are mainly distributed in the southern part of Yantian District, the southwestern part of Longgang District, and the eastern part of Luohu District. The interaction relationship between different urban functional area types during weekdays is

similar to the relationship during weekends. The top three urban functional area types in terms of outflow are IRFA, RCFA, and PCFA, while the top three functional area types in terms of inflow are PRFA, GSFA, and PIFA.

(3) The traffic flow network during weekdays shows a spatial pattern of "dense in the west and south, sparse in the north and west", indicating that there is a significant spatial unevenness in the taxi flow of Shenzhen's residents. Compared with weekdays, the network during weekends is more dispersed, while the differences in node sizes decreased, indicating that travel destinations are more diverse. The taxi mobility network on weekends is more consistent with the characteristics of a small world network than during weekdays. The average degree and the average number of connected edges during the weekday morning peak are higher than during the weekday evening peak, indicating that the influence of commuting behavior is more significant during the morning peak, leading the network interactions to be more frequent. The pattern of community division is more consistent with the administrative division during weekdays, indicating that taxi trips during weekdays are predominantly within the districts. The community characteristic during weekends is clearly different from the community characteristics during weekdays.

(4) Based on our findings, we attempt to provide few policy recommendations. First, in accordance with the distribution of urban functional area units and the characteristics of taxicab trips, we suggest that the transportation service capacity of the outer circle residential areas, such as Guangming District, Pingshan District, and Dapeng New District requires strengthening. The suggestion can provide a reference for taxi companies to develop their operating vehicle deployment plans for different areas and for taxi drivers to develop their cruising plans. Second, based on the characteristics of the mobility network between different UFA units during the morning and evening peak hours, we suggest that housing should be encouraged to be built around employment concentrations, such as higher education institutions, large research institutes, and industrial parks in a mixed land use manner. This can promote the integration of employment space and residential space, thus reducing traffic emissions due to commuting behavior. The suggestion can provide a reference for local authorities to develop urban planning. Third, based on the characteristics of the mobility network between different UFA units, we suggest that the proportion of green and landscaped land within the urban core (Nanshan District, Futian District, and Luohu District) should be increased. This suggestion may help reduce the volume of trips made by people living in the core area to parks and other attractions in the peripheral areas during weekends, and reduce the traffic emissions generated by long-distance trips. The suggestion may provide a reference for the local authorities to develop urban planning. Finally, as well known, the implementation of transit-oriented development (TOD) strategy has become a consensus for urban development in densely populated areas around the world. Therefore, we recommend that the coverage of rail transit stations should be increased to the UFA units with lower mobility index, especially the southeastern part of the city. This proposal can serve as a reference for local authorities and stakeholders involved in the design of new metro lines.

Our findings can prompt the understanding of urban mobility between different UFA units. Compared with other studies in the same area, this paper has some novelty in the setting of UFA as traffic analysis units and the use of complex network methods. For example, Daniel et al. analyzed urban taxi travel behavior in Shenzhen and found that distance, travel time, and road preference have a comparable higher influence on drivers' route choice [60]. Nie examined the impact of ride sourcing on the taxi industry and explored where, when, and how taxis can compete more effectively using a large taxi GPS trajectory data set collected in Shenzhen from January 2013 to November 2015 [31]. Tu et al. explored spatial variations of multi-modal public ridership, such as buses, metro systems, and taxis, and the underlying controlling factors in Shenzhen. In addition, they claimed that employment, mixed land use, and road density have significant effects on the ridership of each mode [61]. Zheng et al. used the taxi trajectory data to investigate the spatial layout and the allocation of management resource of the urban public green space from

the spatial interaction perspective [62]. Shen et al. analyzed the spatiotemporal pattern of taxi travel based on an improved network kernel density estimation method [63]. Gao et al. investigated the impact of the modifiable areal unit problem (MAUP) for understanding the relationships between commuting demand and built environment [64]. Feng et al. explored the spatiotemporal variations of taxi travel routes using taxi data from five large cities (including Shenzhen) and found that human travel can be highly non-homogeneous with power-law scaling distributions of distances and times [65]. In previous studies, the traffic analysis unit is mainly a regular grid, and the irregular polygons obtained using road network segmentation are less studied. Moreover, studies that consider the urban functional attributes of traffic analysis units are especially rare. In addition, the study of spatial interactions between different UFA units by applying complex network analysis methods is relatively limited. Undoubtfully, our study still has some limitations. First, due to the differences between the classification system of Gaode POI and the classification system of urban land use and planning standards of development land, the POI reclassification operation in our study is bound to be somewhat subjective, which may affect the accuracy of the urban functional area identification results. Therefore, remote sensing data, street view data, and fieldwork data should be introduced in future studies to enhance the accuracy of results. Second, the taxi OD data used in this paper only include traditional cruised taxis and does not include the online car-hailing, thus leading to less comprehensive results. In addition, the taxicab trip data in 2017 do not reflect the latest characteristics of urban taxi mobility. Dingil et al. confirmed the reflection of spatial economic inequalities during the COVID-19 pandemic, and demonstrated that travel distance and income level are the two most influential factors in pandemic decision-making [66]. Since the impact of the COVID-19 pandemic on the urban mobility is significant, it is worth noting. In future studies, more diverse and latest trajectories data should be included to reflect urban mobility characteristics in a comprehensive and timely manner. Finally, this paper only analyzes the taxi pick-up and drop-off points, while taxi trajectory data also contain important information regarding travel behavior, which can be applied to human mobility research. Therefore, the inclusion of taxi trajectory data should be considered in the future.

**Author Contributions:** Conceptualization, Guijun Lai and Guanwei Zhao; methodology, Guijun Lai and Guanwei Zhao; software, Guijun Lai, Muzhuang Yang and Guanwei Zhao; validation, Muzhuang Yang, Binbao He and Yuzhen Shang; formal analysis, Guijun Lai and Yuzhen Shang; investigation, Guijun Lai and Yuzhen Shang; resources, Guanwei Zhao and Muzhuang Yang; data curation, Guijun Lai, Binbao He and Guanwei Zhao; writing—original draft preparation, Guijun Lai and Guanwei Zhao; writing—review and editing, Guijun Lai and Guanwei Zhao; visualization, Guijun Lai and Guanwei Zhao; supervision, Guanwei Zhao. All authors have read and agreed to the published version of the manuscript.

**Funding:** This research was funded in part by the Natural Science Foundation of Guangdong Province, China (grant no. 2017A030313240), Philosophy and Social Science Research Program of Guangzhou city, Guangdong Province, China (grant no. 2020GZGJ183), Guangzhou Science and Technology Plan Project—Joint Project Funding by City and University (grant no. 202102010413), and Training Programs of Innovation and Entrepreneurship for Undergraduates in Guangzhou University, Guangdong Province, China (grant no. S202011078001, XJ202111078243).

**Institutional Review Board Statement:** Not applicable.

**Informed Consent Statement:** Not applicable.

**Data Availability Statement:** Not applicable.

**Acknowledgments:** The authors would like to thank the editors and anonymous reviewers for their insightful suggestions and comments.

**Conflicts of Interest:** The authors declare that there is no conflict of interest regarding the publication of this paper. The funders had no role in the design of the study; in the collection, analyses, or interpretation of the data; in the writing of the manuscript, or in the decision to publish the results.

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
