# Peer review of "Revealing Taxi Interaction Network of Urban Functional Area Units in Shenzhen, China"

_ijgi, doi:10.3390/ijgi11070377_

Round 1

Reviewer 1 Report

Review of the manuscript “ Revealing taxi interaction network of urban functional area units in Shenzhen, China”

          Authors applied the complex network analysis method to study the spatial interaction characteristics of taxi flow between different urban functional area units in Shenzhen. The results can provide a better understanding of urban mobility characteristics in Shenzhen, and provide a reference for urban planning and traffic management.

       In general, the work presents a relevant original contribution, has an appropriate research design for the study, the results are very interesting and very useful. The presentation of the methodology used and the results is also quite clear.

       I believe that the work has a very important contribution to the spatial interaction characteristics of taxi flow. So in general I believe that the work is ready for publication, and so my recommendation is for the acceptance of the article in its present form.

     The article fits the scope of International Journal of Geo-Information and will be of interest to its readers.

Reviewer 2 Report

The present study focuses on the identification of urban functional areas in Shenzhen, China by using the taxi mobility dataset. I did not understand half of what I read, the paper should have a language editing service. The abstract must be rewritten by showing the fundamental gain of the study. There is a pleonasm in the introduction and literature review, some things repeat, therefore, I suggest using brief language, and eliminating some parts such as an example lines 42-44 are not necessary. The literature review section is more like methodologic reasoning, you can merge it into the introduction. I suggest creating a new introduction section with the subsections such as background, methodological reasoning, and the aim of the study to increase the traceability of the paper. There are some information and generalizations done without referring, these should be justified by the source or the literature. Some examples are such as line 256: “According to previous studies, 50% was used as the threshold value.’’, which studies?? What is the source of the lines 140-144?? Where is the citation of the trip data and land use data sources?!, so on…. Please read all text, and fix all these types of flaws. The evaluation of the results is vague, please be clearer, and briefer and provide a wiser interpretation. It is unclear what can we do with these results. What is the output of this work? Just re-identification of urban functional areas in Shenzhen? How do you assess the results considering the recent pandemic mobility changes (e.g. https://doi.org/10.1080/19427867.2021.1901011)? Following, how can these results be utilized in terms of sustainable urban transportation & land use planning in Shenzhen?, etc..

Reviewer 3 Report

The article is interesting and the problem is worth exploring, but it is recommended to complement the discussion and conclusion section with a clear indication of what is innovative in the presented solution in comparison to other such solutions (authors' contribution to the development of science). It would also be good to characterize the barriers that the authors faced in their experiments and to point out the advantages of the presented solution compared to other research in this area.

In my opinion, the purpose of conducting the research and research methods are rather entirely clear. However, the conclusions were also easy to predict.

The discussion could be developed by comparing the research with others in the area presented by the authors (the discussion should refer to the purpose of the research ).

The conclusions should state in which scientific and practical areas the presented research can be useful.

In principle, it is difficult to evaluate the suitability of the presented solutions due to the lack of verification with real traffic conditions.

Reviewer 4 Report

Page 1. Keywords: My suggestion is to include “Point-of-Interest, POI” and “Urban Mobility” in the Keywords.

Section 1. Introduction: My suggestion is to substantially increase the text in the specific Section. There are 32 references in the first one and a half paragraphs (pages 1 and 2). I strongly believe that the topics covered by the specific 32 references must be presented with more details within the manuscript since they are of great interest for the reader.

Section 3. Materials and Methods: Please include a Data Flow Chart (DFC) describing all your methodological steps. The proposed DFC will help the reader to obtain a clear overview of your research from the early beginning of the paper.

Subsection 3.1. Study area: My suggestion is to include a detailed description of the taxi system in the city of Shenzhen (e.g., number of taxis, fares etc.). In addition, please include a paragraph concerning the modal split in the city of Shenzhen as well as the traffic and mobility characteristics of the transportation system, for the benefit of the reader.

Page 4, Figure 1. The study area map.: Please include the source of the geographical background (map).

Page 4, line 152, “…The taxis trip data of Shenzhen city was collected in 10, 11, 15 and 16 April, 2017…”: Are there any more recent data available? Did you consider the effect of COVID-19 on mode choice in your analysis?

Page 9, line 339, “The statics of identification…”: Could you please verify that “static” is the correct word?

Page 10, Figure 2. The identification results of urban functional area: Please include the source of the geographical background (map).

Page 12, Figure 5. The map of taxis mobility index during the peak periods: Please include the source of the geographical background (map).

Section 5. Conclusions and Prospect: My suggestion is to add extra text which must be dedicated to the policy recommendations arising from your findings. Please address each one of your recommendations to the respective stakeholder. In addition, please carry out a stakeholders’ analysis to identify who and how will benefit from your findings (e.g., taxi operators, urban traffic control centers, local authorities etc.).

Round 2

Reviewer 2 Report

The revision is satisfactory.

The final language check is required before publishing.

Reviewer 3 Report

I would like to thank the authors for their responses and for improving the paper. I think the paper is suitable for publication.